# AKT3 Expression in Mesenchymal Colorectal Cancer Cells Drives Growth and Is Associated with Epithelial-Mesenchymal Transition

**DOI:** 10.3390/cancers13040801

**Published:** 2021-02-14

**Authors:** Joyce Y. Buikhuisen, Patricia M. Gomez Barila, Arezo Torang, Daniëlle Dekker, Joan H. de Jong, Kate Cameron, Sara Vitale, Giorgio Stassi, Sander R. van Hooff, Mauro A. A. Castro, Louis Vermeulen, Jan Paul Medema

**Affiliations:** 1Laboratory for Experimental Oncology and Radiobiology, Center for Experimental Molecular Medicine, Cancer Center Amsterdam, Amsterdam UMC, University of Amsterdam, 1105 AZ Amsterdam, The Netherlands; j.y.buikhuisen@amsterdamumc.nl (J.Y.B.); p.m.gomezbarila@amsterdamumc.nl (P.M.G.B.); a.torang@amsterdamumc.nl (A.T.); danielle.dekker@amsterdamumc.nl (D.D.); j.h.dejong@amsterdamumc.nl (J.H.d.J.); k.cameron@amsterdamumc.nl (K.C.); S.R.vanHooff-3@prinsesmaximacentrum.nl (S.R.v.H.); l.vermeulen@amsterdamumc.nl (L.V.); 2Oncode Institute, 3521 AL Utrecht, The Netherlands; 3Dipartimento di Medicina e Chirurgia Traslazionale, Istituto di Patologia Generale, Università Cattolica del Sacro Cuore, 20123 Milano, Italy; sara.vitale@unicatt.it; 4Department of Oncology and Molecular Medicine, Istituto Superiore di Sanità, 00161 Rome, Italy; 5Cellular and Molecular Oncology Section; Department of Surgical, Oncological and Stomatological Sciences, University of Palermo, 90127 Palermo, Italy; giorgio.stassi@unipa.it; 6Bioinformatics and Systems Biology Laboratory, Federal University of Paraná, 81520-260 Curitiba, Brazil; mauro.castro@ufpr.br

**Keywords:** mesenchymal CRC, CMS, AKT3, growth

## Abstract

**Simple Summary:**

Colorectal cancer can be subdivided into four distinct subtypes that are characterised by different clinical features and responses to therapies currently used in the clinic to treat this disease. One of those subtypes, called CMS4, is associated with a worse prognosis and poor response to therapies compared to other subtypes. We therefore set out to explore what proteins are differentially expressed and used in CMS4 to find potential new targets for therapy. We found that protein AKT3 is highly expressed in CMS4, and that active AKT3 inhibits a protein that stalls growth of cancer cells (p27^KIP1^). We can target AKT3 with inhibitors which leads to strongly reduced growth of cancer cell lines that are categorised as CMS4. Furthermore, our data suggests that high AKT3 expression in tumour cells may be used to identify poor prognosis colorectal cancer patients. Future research should point out if high AKT3 expression can be used to select colorectal cancer patients that have a poor prognosis but that could benefit from AKT3-targeted treatment.

**Abstract:**

Colorectal cancer (CRC) is a heterogeneous disease that can currently be subdivided into four distinct consensus molecular subtypes (CMS) based on gene expression profiling. The CMS4 subtype is marked by high expression of mesenchymal genes and is associated with a worse overall prognosis compared to other CMSs. Importantly, this subtype responds poorly to the standard therapies currently used to treat CRC. We set out to explore what regulatory signalling networks underlie the CMS4 phenotype of cancer cells, specifically, by analysing which kinases were more highly expressed in this subtype compared to others. We found AKT3 to be expressed in the cancer cell epithelium of CRC specimens, patient derived xenograft (PDX) models and in (primary) cell cultures representing CMS4. Importantly, chemical inhibition or knockout of this gene hampers outgrowth of this subtype, as AKT3 controls expression of the cell cycle regulator p27^KIP1^. Furthermore, high *AKT3* expression was associated with high expression of epithelial-mesenchymal transition (EMT) genes, and this observation could be expanded to cell lines representing other carcinoma types. More importantly, this association allowed for the identification of CRC patients with a high propensity to metastasise and an associated poor prognosis. High *AKT3* expression in the tumour epithelial compartment may thus be used as a surrogate marker for EMT and may allow for a selection of CRC patients that could benefit from AKT3-targeted therapy.

## 1. Introduction

Colorectal cancer (CRC) is a heterogeneous disease. Two major types of CRC can be identified, each characterised by well-defined biological features. The chromosomal instable (CIN) class encompasses the majority of sporadic CRC cases (~85%). Mutations in APC, KRAS, TP53, PIK3CA or PTEN and SMAD4 or SMAD2 frequently occur in these tumours, leading to dysregulation of the Wnt, MAPK, p53, PI3K and TGFβ cell signalling pathways, resulting in enhanced and sustained pro-survival and proliferation signals. The second subgroup comprises hypermutated CRCs characterised by microsatellite instability (MSI). Inactivating mutations or epigenetic silencing of mismatch repair genes and BRAF^V600E^ mutations are abundant in these tumours [1,2]. The existence of these two major classes bears clinical relevance, but their associated mutations or differential activity of cell signalling pathways alone cannot fully explain the distinct prognosis or response to currently used therapies [3].

Recent efforts therefore use transcriptomics-based unsupervised classification to subdivide CRC into clinically relevant groups based on gene expression signatures [4,5,6,7,8,9,10,11]. Certain mutations and clinical features are enriched in some of the four identified consensus molecular subtypes (CMS) and importantly all CMS are characterised by differential activity of signalling pathways involved in distinct biological processes [11]. The CMS4 subtype presents with the poorest clinical outcome and is associated with features such as high stroma infiltration, activated transforming growth factor β (TGFβ) signalling and epithelial-mesenchymal transition (EMT) [11]. Compared to other CMS, CMS4 tumours are more resistant to chemo- and targeted therapies currently used in the clinic [5,6,10,12,13,14,15,16,17,18,19,20]. The aggressive nature of this subtype combined with its refractory response to therapy highlight the need to better understand the biological behaviour that distinguishes it from other subtypes. Elucidating the unique biology of this subtype could aid in designing more effective therapy regimens for this specific CMS.

Stroma-derived transcripts can partially contribute to differential pathway activity between subtypes; Immune cells infiltrate CMS1 tumours more frequently and CMS4 tumours contain relatively high percentages of cancer associated fibroblasts [11,21,22,23]. The challenge is therefore to distil what CMS gene expression patterns can be attributed to the cancer cells specifically. Isella and colleagues have addressed this conundrum using bioinformatical filtering of gene sets and have derived CRC intrinsic subtype (CRIS) classification, also suited for pre-clinical model stratification [24,25]. We and others have shown that by selecting the proper gene sets the same models can be robustly classified according to the CMS system, also across consecutive passages of patient-derived xenografts (PDX) [26,27,28]. These results indicate that CRC cells can preserve their CMS in different microenvironmental settings [27,28,29]. Successful CRIS and CMS classification of pre-clinical models highlights that cancer cell intrinsic differential gene expression patterns exist in CRC that contribute to subtype assignment. However, it does not rule out that crosstalk between stroma and cancer cells shape the instalment of subtype-related gene expression patterns in the epithelial compartment of patients’ tumours.

We set out to identify and study cell signalling pathways specifically active in the epithelial cells of CMS4. We decided to focus our analysis on differential expression of kinases between subtypes, as they represent a class of proteins critically involved in signal transduction pathways that determine cell behaviour. Some chemical inhibitors targeting these proteins are currently successfully used in the treatment of CRC patients. Our approach revealed AKT3 to be expressed in the epithelial fraction of CMS4. This kinase and its closely related family members AKT1 and AKT2, are critical players in the PI3K pathway, and can control metabolism, cell survival and proliferation by regulating targets such as FOXO3a, p27^KIP1^ and TSC2 [30]. The PI3K pathway is frequently activated in the CRC setting, either through mutations in PIK3CA or PTEN [2] or through increased expression and activation of AKT1 and AKT2 during oncogenic transformation [31,32]. Here, we describe that activated AKT3 contributes to the outgrowth of mesenchymal CMS4 cell lines. Furthermore, high *AKT3* expression in CRC cell lines is associated with enriched expression of EMT marker genes, and this strong association can be expanded to a diverse array of carcinoma cell lines.

## 2. Materials and Methods

### 2.1. Gene Expression Datasets and CMS Labels of CRC Patients

Publicly available Affymetrix microarray gene expression datasets of CRC samples were obtained from the Gene Expression Omnibus (GSE33113, GSE39582, GSE35896, GSE13294, GSE14333, GSE17536, GSE20916, GSE2109, GSE23878, GSE37892, and GSE13067) and were complemented with a proprietary dataset (KFSYSCC, made available by Sage Bionetworks through www.synapse.org (accessed on 8 January 2021)) containing 307 CRC samples. Tumour samples present in multiple datasets were removed. To eliminate technical and batch effects caused by the generation of gene expression datasets in different labs and on different platforms, the collected datasets were quantile normalised first using the preprocessCore package in R version 3.6.3 [33], followed by application of the comBat function in the sva package to complete normalisation of the datasets [34]. This dataset aggregate was further complemented with RNAseq data of The Cancer Genome Atlas Research Network which was normalised in the same way [2]. As the platforms used to obtain gene expression data differed from the Affymetrix arrays, org.Hs.eg.db [35], hgu133plus2.db [36] and annotate [37] were used to annotate probe and gene IDs and to properly merge all datasets using the standardised gene IDs before applying the same normalisation steps. Principal component analysis confirmed that normalisation removed dataset clustering.

All datasets have been used before by the CMS classification consortium [11] and CMS labels were directly obtained from Synapse (ID: syn2623706). The final combined dataset contains 2738 samples amongst which 2416 samples are CMS stratified. From this dataset, patients were selected for relapse-free survival analysis using the following criteria: (1) TNM stage I–III; (2) clinically proven microsatellite stable; (3) relapse-free survival data publicly available. These criteria yielded a subset consisting of 376 CRC patients.

### 2.2. Generation of Merged CRC (Primary) Cell Line Dataset

Gene expression data of publicly available cell line datasets (GSE59857, GSE36133, GSE100478) was combined with the data of two public primary cell culture datasets (GSE100549, GSE100479) and an independent RNAseq dataset of 30 primary spheroid cultures derived at the University of Palermo (Giorgio Stassi, Italy) and the Instituto Superiore Sanita in Rome (Sara Vitale, Italy). Combination of these source datasets resulted in a merged dataset containing 292 gene expression profiles, some of which belonging to the same cell line included in multiple source datasets. The same normalisation procedure was used as described above for the CRC patient datasets. Unsupervised hierarchical clustering after normalisation revealed that profiles of the same cell line clustered together. The curated patient and cell line datasets were used as input for the differential expression analysis.

2Log expression levels of individual genes in both patient and cell line samples were obtained from the compiled dataset normalised as a whole. Expression levels for cell lines profiled in multiple datasets were taken from the dataset using the newest, most advanced gene expression profiling platform.

### 2.3. CMS and CRIS Classification of CRC (Primary) Cell Lines

CMS classification was performed on the merged dataset described in Section 2.2. using the support vector machine classifier described in Linnekamp, van Hooff et al. [28]. If multiple gene expression profiles were available for a particular cell line, when they were arrayed in multiple datasets, CMS classification was performed based on each individual profile. Final CMS was assigned based on the following considerations: (a) if a cell line gene expression profile was uniquely available in one source dataset, threshold for the CMS prediction score was set to ≥0.5; (b) cell lines expression profiled in multiple source datasets needed to be consistently CMS classified in ≥66% of the cases with a prediction score >0.4. Using this method, 139 cell lines could be successfully CMS classified Appendix A. CRIS classification was performed on all available gene expression profiles within the same merged dataset using the published CRISclassifier [24] using the default Benamini-Hochberg-corrected false discovery rate (FDR < 0.2). When gene expression profiles of one cell line were present in multiple source datasets, predicted CRIS needed to be concordant in >66% of the dataset to be considered as a confident classification Appendix A.

### 2.4. Kinase Differential Expression Analysis

A comprehensive list including 503 kinases was composed using the InterPro IPR000719 and UniProt release 2016_10 repositories. Gene expression data for these proteins was available for 496 and 490 kinases in the CRC tumour and (primary) cell line datasets, respectively. Kinases differentially expressed between CMS4 and CMS1-3 cell lines and tumours were identified using the limma R package [38] with a threshold for 2log fold change >0.8 and a *p*-value < 0.05.

### 2.5. Gene Set Enrichment Analysis

The merged dataset described in Section 2.2 was used for all gene set enrichment analyses (GSEA) performed on CRC cell lines. For GSEA on a larger panel of cell lines representing multiple tumour types the raw RNAseq counts was obtained from the Cancer Cell Line Encyclopaedia [39] and quantile normalised using the preprocessCore R package [33]. Cut-off for *AKT3* and *FRMD6* high and low expression were set at top and bottom 20% for the CRC cell line panel to enhance the difference and at 50% for the larger pan-cancer cell line panel as well as for the analysis of individual tumour types represented in that panel. The publicly available HALLMARK_EPITHELIAL_MESENCHYMAL_TRANSITION signature was obtained from MSigDB (v7.1) [40]. GSEA were performed and enrichment score and p-values were calculated using the fgsea R package [41].

### 2.6. Cell Culture

Adherent colon cancer cell lines were a kind gift from the Sanger Institute (Cambridge, United Kingdom). CaR-1, CL-40, Gp5d, HT55, Hutu-80, LoVo, LS123, LS180, NCI-H630, OUMS-23, RKO, SW48, SW948, SW1116, SW1417, SW1463 and T84 were maintained in 1:1 DMEM/F12 medium containing 15 mM HEPES and 2.5 mM L-glutamine (cat. #31330095, Gibco, Paisley, Scotland) supplemented with 8% fetal bovine serum (cat. #s-FBS-SA-025, Serana, Pessin, Germany) and 50 units/mL of penicillin/streptomycin (cat. #15140122, Gibco). CCK-81, Colo320-HSR, HCT116, KM12, LS411N, LS513, LS1034, MDST8, NCI-H716, RCM-1, Snu-C1 and Snu-C2B were maintained in RPMI 1640 medium containing 25 mM HEPES and 2.05 mM L-glutamine (cat. #52400041, Gibco) supplemented with 8% fetal bovine serum, 1% D-glucose solution plus (cat. #G8769, Sigma-Aldrich, Saint Louis, MO, USA), 1 mM sodium pyruvate (cat #11360070, Gibco) and 50 units/mL of penicillin/streptomycin. Cells were cultured at 37 °C and 5% CO_2_ in a humidified incubator.

Authenticity of cell lines was confirmed by performing short tandem repeat profiling on 10 ng genomic DNA (isolated with the NucleoSpin^®^ Tissue kit, cat. #740952, Macherey-Nagel, Düren, Germany) using the PowerPlex 16^®^ system (Promega, Madison, WI, USA) according to the manufacturer’s protocol. PCR amplification reaction was run using the following program: 95 °C for 15 min, 96 °C for 1 min; 10 cycles of 94 °C for 30 s, 60 °C for 30 s, 70 °C for 45 s; 20 cycles of 90 °C for 30 s, 60 °C for 30 s, 70 °C for 45 s; and 60 °C for 30 min. Results we analysed using GeneScan^®^ software (Applied Biosystems, Foster City, CA, USA). Cultures were tested for mycoplasma contaminations on a monthly basis.

### 2.7. RNA Isolation and Gene Expression Profiling of PDX Models

Methods with regards to CRC PDX establishment, RNA extraction and CMS classification of these models were described before [28,29]. The same source RNA of a selection of these PDX models was used for additional gene expression profiling by RNA sequencing, performed by BaseClear B.V. (Leiden, The Netherlands) at a read depth of approximately 30 × 10^6^ per sample. All reads were mapped to both the human and murine genome. The alignments were subsequently assigned to either genome using the XenofilteR R package [42]. Gene counts were determined using the summarizeOverlaps function of the GenomicAlignments [43] R package and normalised to FPM (fragments per million mapped fragments) using the DESeq2 R package [44]. Finally, the comBat function in the sva R package was used to remove existing batch effects.

### 2.8. Human Tumour Sample Collection

Human CRC tissue was collected in compliance with Dutch legislation and following written informed consent of patients. The Medical Ethical Committee of the Academic Medical Center (AMC) approved collection of human CRC tissue of patients operated on in the AMC, for research purposes (METC number NL59811.018.16). The bulk of the tumours was processed according to the standard operating procedures in place at the AMC pathology department. Samples from Palermo and Rome were obtained and cultured as described before [45,46] and processed for RNAseq after local ethical approval of both research institutions.

### 2.9. RNA in Situ Hybridisation on Human CRC Tissue

Formalin-fixed, paraffin-embedded (FFPE) tissue blocks for four CRC biobank patients-one MSI, three MSS-were requested from the AMC pathology department and RNA in situ hybridisation was performed on tumour sections using the RNAscope^®^ platform (*AKT3* probe cat. # 434211, RNAScope^®^ 2.5 HD reagent kit brown, cat. #322371, both from Advanced Cell Diagnostics, Newark, CA, USA) according to manufacturer’s protocol. RNAscope^®^ probes recognising bacterial *DAPB* and human *PPIB* we applied to tissue slides as negative and positive controls. Slides were counterstained with haematoxylin and mounted in Pertex before visualisation. Tumour sections were scanned on an IntelliSite Ultra Fast 1.6 slide scanner (Philips, Eindhoven, The Netherlands). Additional RNAScope^®^ was performed on tissue microarrays (TMA) available for the AMC90 patient dataset as described above [47]. A total of 78 TMA cores were available for this dataset, 47 of which passed mRNA quality control based on *PPIB* staining intensity. A core was scored as *AKT3* positive if staining could be observed in the cancer cell fraction of the TMA core.

### 2.10. RNA Isolation, cDNA Conversion and qRT-PCR on Cell Line Samples

Cell lines were grown to reach 70–80% confluency and were harvested using trypsin. Cell pellets were snap frozen in liquid nitrogen and stored at −80 °C until RNA isolation. RNA was isolated using the NucleoSpin RNA kit according to manufacturer’s protocol (cat #740955, Machery-Nagel) and quantified on a NanoDrop 2000 (Thermo Fisher Scientific, Waltham, MA, USA). 1 µg of RNA was converted into cDNA using the Superscript III reverse transcriptase kit according to manufacturer’s protocol (cat. # 18080085, Invitrogen, Waltham, MA, USA). qRT-PCR was performed using SYBR green I (cat. #4887352001, Roche, Basel, Switzerland) on a LightCycler^®^ 480 II machine (Roche). Primer sequences were obtained from PrimerBank [48] or designed in house and are listed in the Table 1 below. All obtained Ct values were normalised to the expression of *ATP5E*, results normalised to *GUSB* were similar.

### 2.11. Immunoprecipitation

Cells were grown to 70–80% confluency on a 15 cm cell culture dish before serum starvation for 6 h and subsequent treatment with 5 ng/mL of human recombinant IGF-1 (cat. #100-11, PeproTech, London, UK) or control for 20 min. All lysis and immunoprecipitation steps were performed on ice or at 4 °C with pre-chilled reagents and reagent tubes. In brief, Plates were washed twice with ice cold PBS and cells were scraped off the plate in 500 µL lysis buffer (300 mM NaCl; 50 mM TRIS base; 5 mM EDTA; 0.02% azide; 1% Triton X-100; pH 7.4 and freshly added 1× protease/phosphatase inhibitor cocktail (cat. #5872, Cell Signaling Technology, Danvers, MA, USA)). Lysates were collected in Eppendorf tubes and incubated on ice for 30 more minutes with regular vortexing in between to complete lysis. Lysates were centrifuged at maximum speed for 15 min to pellet cell debris after which supernatant was collected in a clean tube. 15 µL of lysate was mixed with 2× Laemmli buffer (cat. #1610737, Bio-Rad Laboratories, Hercules, CA, USA) plus 10% β-mercaptoethanol and 1x protease/phosphatase inhibitor cocktail (cat. #5872, Cell Signaling Technology), boiled for 10 min at 95 °C and stored at −20 °C until western blotting, where it served as a protein input control. AKT3 antibody (cat. #3788, Cell Signaling Technologies) was added 1:100 to rest of lysate for IP and mixture was incubated overnight on a rotating device. Antibody plus bound protein was pulled down for 2 h using protein G PLUS-Agarose beads (cat. #sc-2002, Santa Cruz Biotechnology, Dallas, TX, USA). Beads were then washed three times with washing buffer (300 mM NaCl; 50 mM TRIS base; 5 mM EDTA; 0.02% azide and 0.1% Triton X-100 (pH 7.4)) before one final wash with cold PBS. IP proteins were eluted from agarose beads by adding 30 µL of 2× Laemmli buffer containing 10% β-mercaptoethanol and 1x protease/phosphatase inhibitor cocktail to beads and incubating at 95 °C for 10 min with occasional vortexing.

Eluates were split in two and equal fractions were run on separate lanes of the same polyacrylamide gel and processed further for western blotting for phosphorylated AKT and AKT3, as described below. Of note: gels were run until 50 kDa protein ladder marker almost ran off the gel to remove excess background signals from IgG light and heavy chain fragments as much as possible. To allow for antibody reprobing, blots were stripped in between using harsh stripping buffer as described in the Abcam protocol (https://www.abcam.com/ps/pdf/protocols/stripping%20for%20reprobing.pdf, initially accessed on 17 March 2014 and regularly checked for updates afterwards).

### 2.12. Western Blotting

Lysates of the panel of all Sanger cell lines assessed for baseline AKT3 protein expression were generated from cultures grown to 70–80% confluency. Cells were washed twice with ice cold PBS and lysed in 2× Laemmli buffer (cat. #1610737, Bio-Rad Laboratories) containing 0.1 M freshly dissolved DTT and 1x protease/phosphatase inhibitor cocktail (cat. #5872, Cell Signaling Technology) whilst scraping the culture dish. All lysates for other western blots were directly prepared in 2× Laemmli buffer plus 10% β-mercaptoethanol and 1× Halt^TM^ protease/phosphatase inhibitor cocktail (cat. #78440, Thermo Fisher Scientific) and scraped from the culture dish. Samples intended for phospho-AKT (Ser437) blotting were treated with control or AKT inhibitor containing serum-free medium for 6 h and treated with IGF-1 for 15 min before harvest. Lysates were collected in Eppendorf tubes and boiled for 10 min at 95 °C and centrifuged at 4 °C at maximum speed for 10 min to remove remaining cell debris. Protein concentration was determined using the Protein Quantification Assay (cat. #740967, Macherey-Nagel) before loading 20 µg of protein onto 4–15% or 10% Mini-PROTEAN^®^ TGX^TM^ Precast Gels (cat. #4561086 and #4561036, Bio-Rad laboratories). Before proceeding with blotting for AKT3 expression in the whole cell lysate of Sanger lines, the polyacrylamide gel was incubated for 15 min in electrophoresis buffer containing 1% 2,2,2-Trichloroethanol (cat. #T54801, Sigma-Aldrich) to allow for tryptophan visualisation and thereby comparison of amount of protein loaded between cell lines [49]. Gel was imaged using the UV sample tray of a Gel Dox EZ system (Bio-Rad Laboratories). 2,2,2-Trichloroethanol visualisation was not employed for other blots. Protein was transferred to 0.45 µm pore PVDF membrane (cat. #IPVH00010, Merck/Millipore, Burlington, MA, USA) or Whatman^®^ Protran^®^ BA83 nitrocellulose membrane (cat. #WHA104001396, Merck/Millipore) using wet transfer for 1 h at 0.33 A (IP blots) or semi-dry transfer for 1 h at 0.0785 A (AKT3), or to Trans-Blot^®^ Turbo™ RTA Mini PVDF Transfer Kit (cat. #1704272, Bio-Rad Laboratories) (phospho-AKT, p27^KIP1^) using the Trans-Blot Turbo Transfer system (Bio-Rad Laboratories). Membranes were blocked in 5% skim milk or 5% bovine serum albumin in TBS + 0.1% Tween-20 (Sigma-Aldrich) and subsequently probed overnight at 4 °C with primary antibody diluted in blocking buffer under light agitation. Primary antibodies used: phospho-AKT (ser473), 1:1000, cat. #4060; AKT1, 1:2000, cat. #2938; AKT2, 1:2000, cat. #3063; AKT3, 1:1000, cat. #3788, p27^KIP1^, 1:1000, cat. #3686 (all Cell Signaling Technology) and GAPDH, 1:5000, cat. #MAB374/6C5 (Merck/Millipore). The following secondary antibodies were diluted in 5% skim milk or 5% bovine serum albumin in TBS + 0.1% Tween-20: goat-α-rabbit-IgG-HRP, 1:5000, cat. #7074 (Cell Signaling Technologies) and goat-α-mouse-IgG-HRP, 1:5000, cat. #1031-05 (SouthernBiotech, Birmingham, AL, USA). Blots were imaged using Lumi-Light^PLUS^ western blotting substrate (cat. #12015196001, Roche) on an ImageQuant^TM^ LAS 4000 machine (GE Healthcare, Chicago, IL, USA). All in between washing steps were performed with TBS + 0.1% Tween-20.

Densitometry readings were obtained by first converting the tiff original western blot image file to a JPEG grayscale format and were imported to Image J. A region of interest (ROI) rectangle was defined and used for each lane to select the band of interest. The mean gray value of the ROI was measured for each band as well as the background region of each lane. The inverted pixel density value was calculated by subtracting 255 from the measurement value obtained for both the protein band and the background region. The net value for the specific protein band was calculated by subtracting the inverted pixel density value of the protein region from the inverted pixel density value of the background region. The same quantification protocol was also performed for the GAPDH western blots and used as a loading control measurement. Finally, the relative quantification value was obtained from the ratio of the net protein value over the net loading control value. All values corresponding to experimental set-up were normalised to control condition to express protein quantity relative to control. All whole western blots can be found in the Appendix A.

### 2.13. Cell Viability Assays

Cell lines were seeded in 100 µL medium in a 96 well cell tissue culture plate and treated in triplicate with DMSO or a titration of AKT1/2 kinase inhibitor (cat. #A6730, Sigma-Aldrich) and MK2206 2HCl (cat. #S1078, SelleckChem, München, Germany), printed using a HP D300e Digital Dispenser (Hewlett-Packard, Palo Alto, CA, USA). Number of cells plated per well: HT55 4000, Snu-c1 2000, Colo320-HSR 1500, Hutu-80 750, LS123 3000, OUMS-23 3000. CellTiter-Blue^®^ Cell Viability Assay (cat. #G8082, Promega) was used as a read-out five days after exposure to treatment, measured on a Synergy^TM^ HT multi-detection microplate reader (BioTek Instruments, Winooski, VT, USA). Blank signal of medium only control was deducted from all reads before normalising all values to the average value of the DMSO treated control (= value treated replicate/average value DMSO triplicate). The resulting normalised value was then subtracted from 1 to acquire the relative decrease in signal compared to the DMSO control.

Proliferative capacity of AKT3 knockout clones compared to wildtype cell lines was determined by plating 5 × 10^4^ cells (Colo320-HSR) and 2 × 10^4^ cells (HuTu-80) in triplicate per condition in a 6 well plate. Cells were harvested using trypsin and live cells were counted in a Bürker chamber using trypan blue 96 h post plating. Experiment was performed at an *n* = 3.

### 2.14. Generation of Stable CRISPR-Edited AKT3 Knockout Clones

Colo320-HSR and Hutu-80 early passage wildtype cell lines were co-transfected with the pX330-hSpCas9 plasmid (cat. #42230, Addgene, Watertown, MA, USA) containing an *AKT3*-targeting gRNA and pcDNA3-eGFP using Lipofectamine 2000 Transfection Reagent (cat. #11668019, Invitrogen). *AKT3* gRNA sequence: 5′–TAAGGTAAATCCACATCTTG. 48 h post transfection, GFP positive cells were single cell sorted into 96 well plates on a 4 laser FACSAria^TM^ II SORP (BD Biosciences, Franklin Lakes, NJ, USA). Successful knockout of AKT3 in single cell clones was assessed by Sanger sequencing (forward primer 5′–cagcattgcaaaaaggttatttt, reverse primer 5′–aacctctatgctaagggactgaaa) followed by indel analysis using TIDE [50], and confirmed on western blot.

## 3. Results

### 3.1. AKT3 is Differentially Expressed between CMS in Tumours and Preclinical Models

As a starting point, we used a large gene expression dataset of colorectal tumours made publicly available by the CMS consortium containing 2416 samples that can be faithfully assigned to one of the four CMSs [11]. We subsequently employed a comprehensive list of kinases to determine if higher expression existed for some of these genes in CMS4 versus other subtypes, as our aim was to find kinases exerting a specific function in this mesenchymal subtype (Figure 1a). Amongst the genes identified were kinases that can be associated with the stromal infiltration and response to extracellular matrix deposition, such as DDR2, FGFR1, PDGFR and CSF1R, and also kinases involved in changes in the actin skeleton and cell shape, migration and EMT and stem cell-like potential such as MYLK, PRKD1, AXL, NUAK1 and DCLK1. Finally, the differential expression of LATS2, a known YAP pathway regulator, and AKT3, point to a difference in proliferation and survival of cells.

It is important to realise though that the relatively high expression of these kinases in CMS4 could reflect the overall enriched association of pathways involved in stromal infiltration previously reported to exist in this subtype [11]. We were however interested in identifying cancer cell specific features and therefore determined whether the differential kinase expression is a result of differential biology in the cancer cells. Therefore, the differential expression analysis was repeated on a curated dataset harbouring 292 gene expression signatures of 220 unique in vitro CRC cultures, as some cultures were profiled across multiple datasets. This set contained both classical cancer cell lines, primary suspension and organoid cultures. A total of 139 unique cell lines could be faithfully CMS classified Appendix A. These cultures solely consist of cancer cells and can therefore be used as a reference to allow for selection of kinases differentially expressed in the epithelial fraction, specifically. Of the 16 kinases originally identified in the tumour dataset, 6 were also found to be differentially expressed in the cell line panel: *AKT3*, *MYLK*, *PRKD1*, *AXL*, *NUAK1* and *DCLK1* (Figure 1a,b). Closer examination revealed that *AKT3* ranked in the top part of both analyses (number 3 and number 2 in the tumour and cell line datasets, respectively), and importantly appeared to be strongly expressed in the majority of CMS4 cell lines and not at all or very lowly in CMS1-3 cell lines, whereas the other 5 kinases were also expressed in cultures assigned to a different CMS (Figure 1c,d). Intriguingly, the differential expression of *AKT3* was unique as its two closely related AKT family members, *AKT1* and *AKT2*, were found to be similarly expressed across all CMSs in cell lines and tumours (Appendix A).

*AKT3* has been categorised before as a kinase that is differentially expressed between subtypes in CRC samples, but this was previously attributed to its high expression in the stromal compartment [21,22]. We thus set out to independently validate our observations in the cell line gene expression dataset. Quantitative PCR on a subset of cell lines corroborated that *AKT3* is solely expressed in CMS4 cell lines whereas detection of *AKT3* in CMS1-3 cell lines bordered the minimum detection range (Figure 2a). This observation was reflected in the lack of AKT3 protein expression in the same CMS1-3 cell lines, while the abundance of *AKT3* mRNA in CMS4 cell lines was clearly translated into protein (Figure 2b). Importantly, expression is not observed in all CMS4 cell lines and a clear range from negative to very high expression within the CMS4 subset exists. On an individual CMS4 cell line level, the relative *AKT3* mRNA expression levels did match those on the protein level. Subtype-specific AKT3 expression could not simply be attributed to a selective mutational profile present in the cell lines, as for instance Colo320-HSR carries similar CRC driver mutations as the AKT3-negative CMS2 lines HT55 and NCI-H630 [28]. Next to cell lines, clinically relevant samples were analysed for *AKT3* expression. First, *AKT3* expression was assessed in an RNAseq dataset of PDXs. As the tumour epithelium in these models is human derived and the stroma is of murine origin, expression can be determined in these separate compartments by dichotomising the species-specific transcripts. This separation confirmed high overall expression of *AKT3* in the stromal compartment across all subtypes, but also revealed that expression of *AKT3* was detected in the epithelial compartment of PDXs that were previously classified as CMS4 (Figure 2c). Finally, the cancer cell selective expression of *AKT3* in tumour resection material was confirmed using RNAscope analysis on tissue slides (Figure 2d) and on TMA cores available for the AMC90 patient dataset [47]. *AKT3* expression was detected in five out of 47 evaluable TMA cores. These results confirm that *AKT3* mRNA is indeed expressed in the stroma of tumours, but in some individual cases also in cancer cells of tumour specimens.

Taken together, these results highlight that AKT3 is a kinase that is highly expressed in the stromal compartment as well as in cancer cells that belong to CMS4, both in preclinical models and patient samples.

### 3.2. AKT3 is Active in CMS4 Lines and Contributes to Outgrowth by Inhibiting p27^KIP1^

In order to corroborate that the increased expression of AKT3 was also linked to activity of this kinase, we set out to address if AKT3 was activated by phosphorylation in CMS4 cell lines. Cell lines were serum-starved to abolish base-line PI3K signalling and subsequently stimulated with insulin-like growth factor 1 (IGF-1), a well-known activator of this pathway [30]. As a specific phospho-AKT3 antibody was not available, we performed immunoprecipitation (IP) for AKT3 and immunoblotted for activated, serine 473 phosphorylated AKT. This method clearly showed that AKT3 is phosphorylated in CMS4 cell lines upon IGF-1 stimulation, in contrast to the CMS2 cell line HT55 in which phospho-AKT3 was not detected, in line with the absence of AKT3 in this cell line altogether (Figure 2b and Figure 3a).

To establish the functional impact of this differential AKT activation two selective AKT small molecule inhibitors were used; one solely targeting AKT1 and AKT2 (AKT1/2 kinase inhibitor) and one targeting all isoforms (AKT1/2/3 inhibitor-MK2206). The AKT1/2 kinase inhibitor as well as the AKT1/2/3 inhibitor were capable of blocking phosphorylated AKT in CMS2 cell lines, indicating that the phospho-AKT signal in control-treated cells represents activation of AKT1 and/or AKT2, concordant with the absence of AKT3 expression in these lines (Figure 3b). As a consequence, CMS2 lines responded in a dose-dependent manner to the AKT1/2 inhibitor when analysing cellular growth, whereas additional targeting of AKT3 by MK2206 did not majorly increase the impact on CMS2 viability (Figure 3c). Treating CMS4 cell lines expressing AKT3 with AKT1/2 kinase inhibitor only slightly reduced phosphorylation of AKT, indicating that CMS4 cells preserved active, phosphorylated AKT3 in this setting. This AKT3 phosphorylation was however significantly decreased by MK2206 (Figure 3d). The lack of effective inhibition of AKT phosphorylation and hence activation by the AKT1/2 inhibitor explains why AKT3 positive CMS4 cell lines were largely insensitive to this drug, whilst viability was decreased in a dose-dependent manner when all three AKT isoforms were targeted by MK2206 (Figure 3e). The differential sensitivity to both drugs is dependent on AKT3 expression rather than on a different CMS4-associated feature, as CMS4 lines that express very low levels of *AKT3* respond equally well to both inhibitors, indicating that a threshold level of AKT3 expression is required for cells to become resistant to AKT1/2 inhibition (Appendix A). These results underline the importance of AKT3 in the viability of CMS4 tumour cells expressing this kinase and we conclude that this AKT family member is providing a survival or proliferation signal, even in the absence of AKT1 and AKT2 activity.

Although the differential sensitivity to the inhibitors shows that AKT3 activity alone is sufficient to sustain outgrowth of CMS4 lines, it does not prove that it is also uniquely required. To determine whether targeting of just AKT3 in CMS4 cell lines would alter their outgrowth, AKT3 knockout single cell clones were generated for Colo320-HSR and Hutu-80 using CRISPR-Cas9. Knockout of AKT3 was confirmed on DNA and protein level (Appendix A). Single cell AKT3 knockout cultures for both cell lines could be expanded and propagated, indicating that loss of AKT3 does not fully hamper proliferation or directly causes cell death. Expansion of all our knockout clones was however significantly impaired in culture compared to their wildtype counterparts (Figure 3f). We wanted to assess if this effect was in part elicited by regulation of the classical downstream AKT3 target p27^KIP1^, a cell cycle regulator of which transcription is inhibited when AKT is active [30]. Indeed, AKT3 knockout cell lines showed an increased expression of p27^KIP1^, which can in part explain the diminished outgrowth of CMS4 AKT3 knockout clones (Figure 3g). This suggests that AKT3 contributes to the growth of mesenchymal/CMS4 cancer cells and that selective inhibition of this kinase could represent a novel therapeutic avenue to decrease growth of this subtype.

### 3.3. High AKT3 Expression Is Associated with Enriched EMT Gene Expression

Previously, *AKT3* expression has been linked to tumours of a mesenchymal origin [51], or to breast cancer cell lines adhering to a more mesenchymal phenotype [52]. This association was therefore analysed in our panel of CRC cell lines. Gene set enrichment analysis revealed a strong expression of genes associated with EMT in CRC cell lines highly expressing *AKT3* as compared to those lowly expressing this kinase (Figure 4a,b). Importantly, this was not solely due to the significant presence of CMS4 cell lines in the *AKT3*-high sample set, which are overall characterised by higher EMT pathway activity compared to other CMS [28], as a similar strong association existed between an EMT signature and *AKT3* expression within the CMS4 cell lines (Figure 4c). This association was equally present in the clinically relevant CRC PDX panel (Appendix A). Intriguingly, we could extend this observation to the CCLE dataset containing gene expression profiles of a wide range of cancer cell lines. When cell lines derived from different carcinomas that all heterogeneously express *AKT3* were analysed, high *AKT3* expression was very strongly associated with EMT gene set enrichment (Figure 4d). Separate analysis of the different cancer types confirmed that this association is also present within each individual tumour types (Appendix A). Specific analysis of a subset of genes known to regulate EMT (*ZEB1/2*, *SNAI2*) or frequently used as markers for EMT (*VIM*, *CDH1/2*) further corroborated the strong association between *AKT3* expression and this program (Figure 4e). Although these results do not necessarily suggest that AKT3 expression directly contributes to or drives the instalment of the mesenchymal phenotype, they do indicate that high epithelial *AKT3* gene expression can be used as a surrogate marker to identify carcinoma cells that have undergone EMT and/or adhere to a more mesenchymal phenotype.

### 3.4. AKT3 and FRMD6 as Marker Genes for Prognosis

EMT is classically associated with metastatic spreading and poor prognosis. To determine if *AKT3* expression could be used to identify patients at risk of recurrence we analysed the expression of *AKT3* in a publicly available cohort of colorectal cancer patients with microsatellite stable local disease (Stage I–III). Comparing patients with the highest *AKT3* expression (last quartile) to the 75% low expressors revealed a trend towards worse relapse-free survival (Figure 4f). This analysis is however strongly confounded by the fact that high expression of *AKT3* is not a sole reflection of tumours that have activated an EMT programme but contains tumours rich in stromal fibroblasts expressing *AKT3* as well. To circumvent this confounder the cancer cell line gene expression patterns were analysed to find epithelial-specific genes of which expression strongly associated with *AKT3* and EMT (Appendix A). Amongst the most significantly correlating genes was *VIM*, a gene generally associated with EMT but also highly expressed in stromal cells and therefore unsuitable for our strategy. In the top-ranking associated genes one gene stood out as it is one of the top centroid genes for a poor prognosis mesenchymal subtype of CRC [7] and was previously shown to be selectively expressed within the epithelial compartment of CRC [14,53]. This gene, *FRMD6*, showed a highly significant correlation with *AKT3* expression and as expected with EMT signatures in the cell line panel (Figure 4g,h and Appendix A). It is important to note that AKT3 does not directly regulate the expression of FRMD6; FRMD6 expression is not downregulated in AKT3 KO cells compared to wildtype cell lines (Appendix A). FRMD6 thus serves as a surrogate epithelial marker for EMT and was therefore used to segregate the same CRC patient set into high and low expressors. Importantly, patients with high *FRMD6* expressing cancers had a significantly higher chance to develop recurrences (Figure 4i). Combined, our data indicates that *AKT3* and *FRMD6* expression strongly associate with EMT and that *FRMD6* can be used as a surrogate marker for EMT and epithelial *AKT3* expression levels to predict prognosis in CRC patients.

## 4. Discussions

Our results establish that AKT3 is expressed in the epithelial, cancer cell compartment of CMS4 CRC in human tumours and in various preclinical models, ranging from PDX models to cell lines, representing this particular subtype. By making use of these models we were able to demonstrate that this kinase is not only differentially expressed in CMS4 but is also activated upon upstream stimulation of the pathway. Loss of this kinase by chemical inhibition or by selective knock out of AKT3 impairs expansion of cell lines where it is normally expressed. The closely related isoforms AKT1 and AKT2 are evidently co-expressed with AKT3, but they cannot fully compensate for loss of AKT3, suggesting that either AKT3 is preferentially used by these cell lines to relay upstream signals or that AKT3 fulfils a unique role in CMS4 mesenchymal cancer cells that cannot be compensated for by AKT1/2.

Our data reveal an intriguing difference between the expression patterns and activity of AKT isoforms in cancer cells. How different AKT isoforms exactly fulfil specific functions and elicit different downstream effects is an ongoing field of research. It has been described that the three isoforms do serve a different physiological role as is evident by the differences in phenotype in mice lacking one of the AKT family members [54,55,56,57,58]. Whether these differences in phenotype relate to organ-specific expression patterns of the respective family members or to differential signalling is less clear. With regards to the latter topic, it is important to note that most AKT substrates can be regulated by all three family members, like p27^KIP1^ reported on in this manuscript, although some isoform selectivity has been distinguished using phospho-proteomics approaches. Localisation of isoforms in different sub-compartments of the cell can influence signalling outcome, as can the relative expression of different AKT members [30,59,60].

Specific roles of respective AKT family members have been reported in different cancer settings, and this has been most consistently described for AKT1 and AKT2 in breast cancer; AKT1 is mainly involved in growth and inhibits migration and metastasis, whereas AKT2 activity is not necessarily linked to proliferation but rather drives the invasive and metastatic capacity of different breast cancer models [30,52,61,62,63,64]. The role of both isoforms is less well studied in CRC and the picture is less clear-cut and dependent on the model system used; AKT2 seems to be similarly important for metastasis formation by CRC cell lines [65,66,67] and a recent study using genetically engineered mouse models reported that combining TP53 inactivation with expression of constitutively active AKT1^E17K^ and azoxymethane treatment gives rise to aggressive, metastatic tumours resembling the mesenchymal CMS4 subtype [68]. Overall, AKT3 is the least studied compared to the other two isoforms. As with the two other isoforms, AKT3 is not often reported to be activated through genetic events; infrequent *AKT3* activating E17K mutations have been identified in melanoma samples. Similarly, a rare *MAGI3-AKT3* activating fusion has been detected in triple negative breast cancer (TNBC) [69,70,71,72]. Like we have seen in our CMS4 CRC samples, AKT3 activity is more often induced by (over)expression of this gene in a particular group of tumours such as basal-like breast cancer or TNBC, serous ovarian cancer and melanoma [51,52,73].

AKT3 has been described to be the AKT isoform predominantly upregulated in melanoma to guide full oncogenic transformation; the consequential activation of PI3K signalling allows for evasion of cell cycle arrest caused by *BRAF^V600E^* oncogene induced senescence [51,74]. The expression, activity and function of AKT3 in controlling proteins involved in cell cycle progression that we describe here in mesenchymal CRC cell lines is, however, more closely related to what has been observed before in TNBC cell lines [52]. Although we did not further elaborate on the role of AKT1 and AKT2 in CMS4, we do find a clear impact of AKT3 loss on proliferation in spite of the two other family members still being expressed, as is the case in TNBC cell lines [52].

One thing these cancers of different origin have in common, aside from high *AKT3* expression, is an association with mesenchymal features. The GSEA shown here provide evidence that the expression of *AKT3* in the epithelial cancer cells is strongly associated with EMT features in CRC cell lines as well as a wide range of cell lines derived from other cancer types, including breast cancer, further hinting at the possibility that the induction of AKT3 expression is part of the instalment of a more mesenchymal cancer cell phenotype. Loss of AKT3 through genetic knock-out did not affect the expression of various EMT marker genes and did not change expression of *FRMD6*. We therefore do not conclude from our data that AKT3 or FRMD6 expression and activity are driving the acquisition of mesenchymal features, but rather see both as candidate markers for presence of EMT. In this light, the earlier reported expression of *AKT3* in the mesenchymal fibroblast compartment in CRC only provides further support for a strong association between *AKT3* and a mesenchymal transcriptional program [5,21,22].

Our data therefore highlight the danger of stringent bioinformatic filtering approaches aimed at excluding stroma-derived signatures from the analysis, as they can lead to loss of important biological insight when gene expression is shared between stromal and cancer cells. This is likely the reason why *AKT3* expression was never linked to a particular CRIS class, even though that classification system specifically aims at describing pathways intrinsically active in cancer cells. Also in this case, filtering against stromal genes would have excluded *AKT3* from the CRIS stratification signature and from further investigation into its role in the epithelium of CRC [24].

Based on our data, it is worth to explore the use of high AKT3 or strongly correlated FRMD6 expression in the epithelial compartment as a surrogate marker for poor prognosis CRC that has undergone an epithelial to mesenchymal transition. We have shown that in situ RNA hybridisation for *AKT3* on FFPE tumour tissue can be used to identify tumours in which *AKT3* is highly expressed in the cancer cell compartment. Alternatively, previously optimised immunohistochemistry for FRMD6 on FFPE material [14] can function as a clinically relevant approach to identify similar patients presenting with poor prognosis mesenchymal CRC that could benefit from treatment with an inhibitor targeting all AKT family members. Our in vitro data as well as that of others indicate that growth of mesenchymal cancer cell lines can be diminished as long as AKT3 is also targeted [52]. Given that outgrowth of CRC cell lines expressing AKT3 was also reduced when only AKT3 was knocked out, it is of interest to aim for development of a more selective AKT3 inhibitor. Such a directed compound could circumvent unwanted effects as a consequence of targeting the functionality of all AKT isoforms, thereby reducing overall toxicity. The use of such a compound may ultimately impact on poor prognosis CMS4 CRC in two different ways: by targeting the cancer cells on one hand, and the abundant supportive fibroblasts on the other.

## Figures and Tables

**Figure 1 cancers-13-00801-f001:**
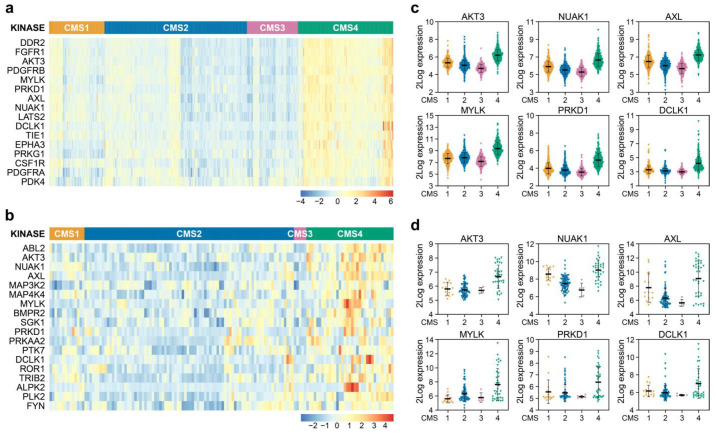
Identification of differentially expressed kinases in CMS4 versus CMS1-3 samples. (**a,b**) Heatmaps depicting Z-score transformed 2Log expression of kinases significantly differentially expressed in CRC patient samples (**a**) and (primary) cell lines (**b**); (**c,d**) 2Log mRNA expression levels of kinases differentially expressed between subtypes in both patient samples (**c**) and (primary) cell lines (**d**) as determined by microarray or RNAseq.

**Figure 2 cancers-13-00801-f002:**
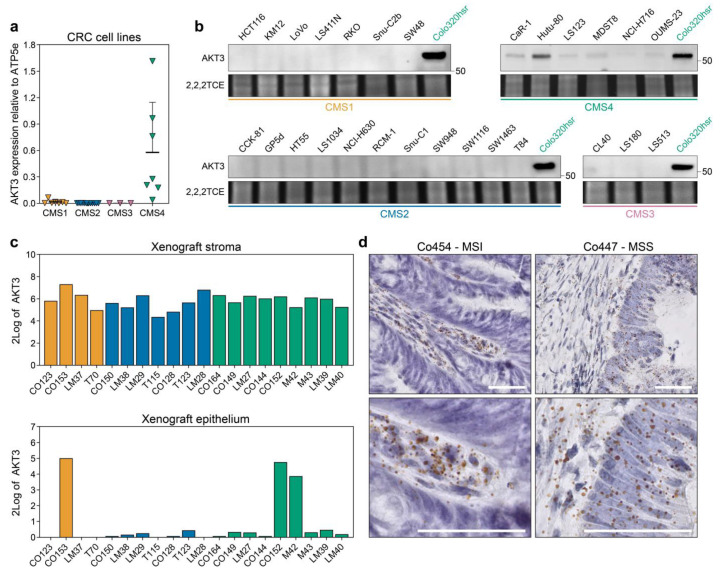
AKT3 is more highly expressed in CMS4 human CRC cell lines and tumours. (**a**) Quantitative real time PCR analysis on 29 established CRC cancer cell lines for *AKT3* expression. Expression level was normalised to *ATP5E* expression, average normalised expression of *N* = 3, each performed with technical triplicates, is plotted per cell line. Normalisation to *GUSB* yielded similar results; (**b**) Western blot analysis for AKT3 expression in the same cell line panel as in a. The CMS4 cell line Colo320-HSR was included on every blot as a positive control. Numbers on right hand side of blots indicate position of specified protein molecular weight marker. 2,2,2-Trichloroethanol (2,2,2TCE) signal (excerpt taken around 60 kDa region) indicates amount of protein loaded per cell line; (**c**) 2Log expression of *AKT3* in the murine stroma and human cancer cells in PDX models as determined by RNAseq; (**d**) Representative images of RNAscope staining for *AKT3* (brown dots) on human CRC tissue slides. Sections were counterstained with haematoxylin. *AKT3* expression was only detected in the stroma of Co454, whereas the epithelium of Co447 also stained positive. MSI, microsatellite instable; MSS, microsatellite stable. Scale bars 50 µm.

**Figure 3 cancers-13-00801-f003:**
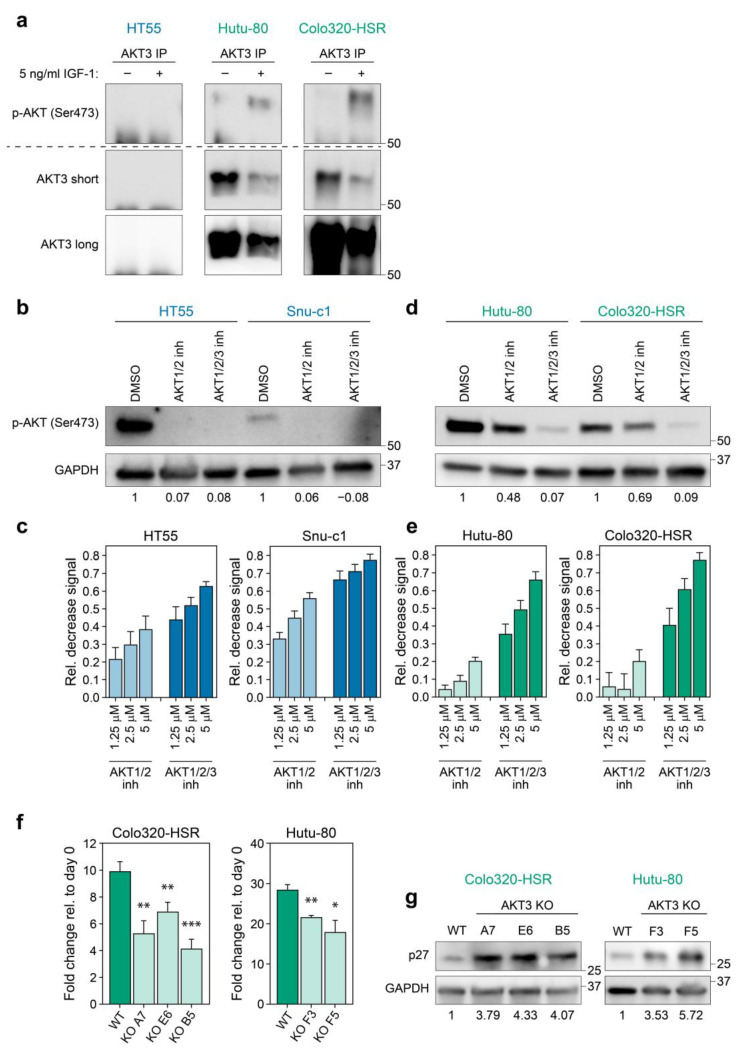
AKT3 is activated in and contributes to growth of CMS4 cell lines. (**a**) Images of western blots performed on AKT3 IP samples of HT55 (CMS2), Colo320-HSR and Hutu-80 (CMS4). Two different exposures are shown for the anti-AKT3 western blots (short = middle panel) and long (bottom panel). Of note IP lysates of each cell line were equally divided over two lanes and then run and blotted on the same gel and membrane. Membrane was cut into two and parts were either probed for serine 473 phosphorylated AKT or AKT3. Signal at the bottom of blots is specific signal coming from the IgG heavy chain. Representative blots of one experiment are shown (total *N* ≥ 3); (**b,d**) Western blot for phospho-AKT (Ser473) in cells treated with 1.25 µM of indicated inhibitor. Numbers at bottom of blots indicate phospho-AKT signal relative to control condition, as calculated after densitometric analysis of bands; (**c,e**) Bar graphs depicting the average decrease in CellTiter Blue signal of CMS2 (**c**) and CMS4 (**e**) cell lines upon treatment with AKT1/2 kinase inhibitor or AKT1/2/3 inhibitor (MK2206). All values were normalised to signal in DMSO treated control conditions. Bar graphs indicate mean of three independent experiments, error bars indicate S.D.; (**f**) Bar graphs depict outgrowth of polyclonal WT population and AKT3 KO clones relative to initial number of cells plated. One representative experiment is shown for 3 independent experiments. Error bars represent S.D. *p*-values calculated using two-tailed Welch’s t-test. * *p*-value < 0.05; ** *p*-value < 0.01; *** *p*-value < 0.001; (**g**) Western blot analysis for p27^KIP1^ expression in polyclonal WT population versus AKT3 KO clones. Blots shown are representative for 2 independent experiments. Numbers at bottom of blots indicate p27^KIP1^ signal relative to control condition, as calculated after densitometric analysis of bands. Numbers on right hand side of all blots in this figure indicate position of specified protein molecular weight marker.

**Figure 4 cancers-13-00801-f004:**
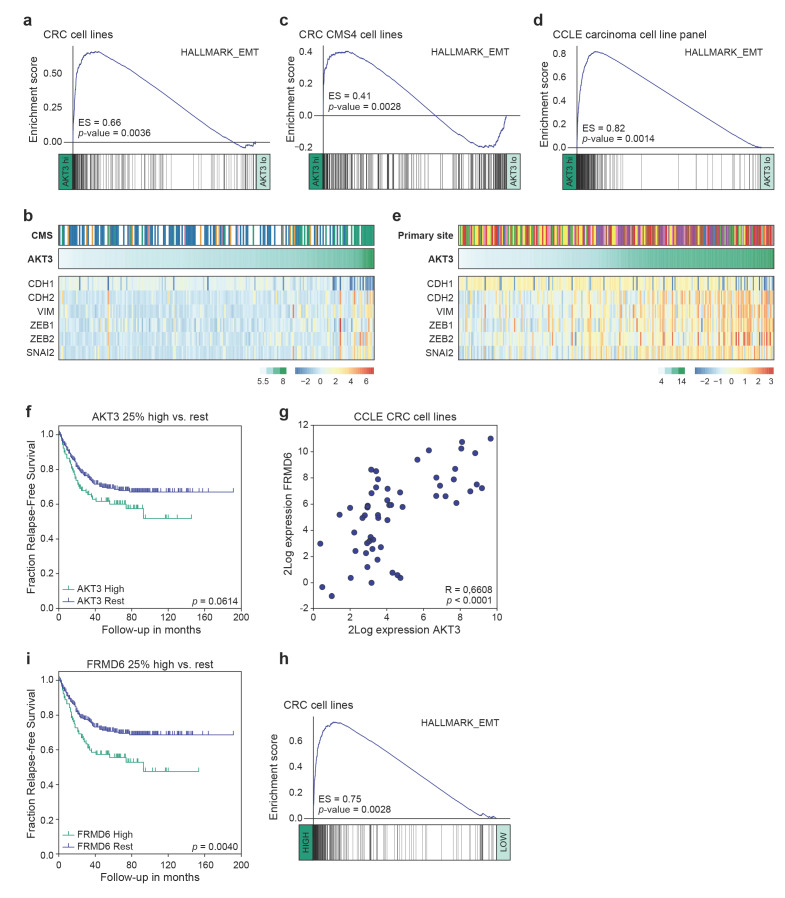
High *AKT3* and *FRMD6* expression levels are linked to EMT gene set enrichment and relapse-free survival. (**a,c**) Gene set enrichment analysis (GSEA) on 20% high versus 20% low *AKT3* expressing CRC cell lines (**a**) and CMS4 CRC cell lines (**c**); (**b**) Heatmap depicting Z-score transformed 2Log expression levels of EMT genes in CRC cell lines. CMS1: orange, CMS2: blue, CMS3: pink, CMS4: green, unclassified: white. 2Log *AKT3* expression levels are indicated on top of heatmap in green gradient; (**d**) GSEA on 50% high versus 50% low *AKT3* expressing carcinoma cell lines included in the CCLE panel; (**e**) Heatmap depicting Z-score transformed 2Log expression levels of EMT genes in panel of CCLE carcinoma cell lines. Colours on top indicate carcinoma of origin; biliary tract: grey, breast: red, endometrium: pink, large intestine: yellow, oesophagus: brown, pancreas: purple, prostate: blue, small intestine: orange, stomach: green. 2Log AKT3 expression levels are indicated on top of heatmap in green gradient; (**f**) Kaplan-Meier curves for relapse-free survival of 376 MSS stage I-III CRC patients present in the CMS consortium dataset, for whom clinical outcome data is publicly available. Last quartile (top 25%) of patients most highly expressing *AKT3* were compared rest of patients in dataset. P-value calculated using Mantel-Cox test; (**g**) Scatter plot depicts 2Log expression of *AKT3* and *FRMD6* in each sample of CCLE CRC cell line panel. R- and p-value calculated using two-tailed Pearson correlation; (**h**) GSEA on 20% high versus 20% low *FRMD6* expressing CRC cell lines; (**i**) Kaplan-Meier curves for relapse-free survival on same patient dataset as in (**f**). Last quartile (top 25%) of patients most highly expressing *FRMD6* were compared to rest of patients in dataset. *p*-value calculated using Mantel-Cox test.

**Table 1 cancers-13-00801-t001:** Primer sequences for quantitative real-time PCR.

Gene	Forward	Reverse
*AKT1*	5′–agcgacgtggctattgtgaa	5′–gcgccacagagaagttgttg
*AKT2*	5′–catcaaagaaggctggctcc	5′–tcttcagcaggaagtaccgtg
*AKT3*	5′–aatggacagaagctatccaggc	5′–tgatgggttgtagaggcatcc
*FRMD6*	5′–ggacactctgggttgattgtg	5′–tgtgtcgatcagtggaggtc
*ATP5E*	5′–ccgctttcgctacagcat	5′–tgggagtatcggatgtagctg
*GUSB*	5′–tggttggagagctcatttgga	5′–gcactctcgtcggtgactgtt

## Data Availability

Links and references to publicly available datasets used for this study are specified in the materials and methods section.

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
