# Peer review of "AKT3 Expression in Mesenchymal Colorectal Cancer Cells Drives Growth and Is Associated with Epithelial-Mesenchymal Transition"

_cancers, 2021, doi:10.3390/cancers13040801_

Round 1

Reviewer 1 Report

Comments on manuscript Cancers-1087859 by Buikhuisen et al.

The manuscript by Buikhuisen et al. is centred on the overexpression of AKT3 in the mesenchymal subtype of colorectal cancer (CRC), its impact on cell growth, its relationship with epithelial-to-mesenchymal transition (EMT), as well as its impact on survival in patients.

The strategy adopted by the authors is to evaluate the comprehensive profile of kinases transcriptional expression in large panels of patient samples and cells lines, as a function of the consensus molecular classification. Their aim is to identify kinases with increased expression in the poor-prognosis CMS4 subgroup, associated with a mesenchymal phenotype. Among the various candidates, they select AKT3 and provide evidence that the AKT3 protein is indeed overexpressed in cells lines, primary cell cultures and patient-derived xenografts (PDX) of the CMS4 subtype, although not systematically. Since AKT3 can be both of tumour (epithelial) and stromal origin, they further evaluated the distribution of AKT3 mRNA in CRC tumours through RNAscope and found that AKT3 mRNAs are upregulated in epithelial cells in some tumours, beyond stromal cells. In a next step, the authors show that AKT3 is activated (phosphorylated) in two CMS4 cell lines and that it negatively controls the expression of the cell cycle regulator p27KIP. The authors further provide evidence for a correlation between AKT3 expression and hallmarks of EMT, not only in CRC cell lines but also in the whole cell panel of the Cancer Cell Line Encyclopaedia. Finally, they show using patient datasets that high AKT3 gene expression, as well as that of FRMD6, a gene that is both correlated with AKT3 and more abundantly expressed in epithelial than in stromal cells is associated with poor prognosis. On this basis, they claim that AKT3 and FRMD6 may help predict prognosis in CRC.

The hypothesis behind the work is well explained and relevant. This study is globally well conducted, the data are well presented, and based on adequate methods. I believe it will be of broad interest to both researchers and clinicians in the field.

However, some major points need to be addressed to fully support the conclusions.

My major concern is that the link between increased AKT3 and the CMS4 subtype is far from absolute. In figure 2B, the authors show that among 7 CMS4 CRC cell lines, 2 express very abundant AKT3 protein levels, 2 express moderate levels and the 3 remaining cell lines appear to be negative. In the same vein, among 9 CMS4 PDX studied, only 2 display high levels of AKT3 mRNA, while one out of 4 CMS1 PDX display high levels of AKT3 mRNA. This point needs to be clarified. Is there any relationship between AKT3 expression and the mutational status of the cells / PDXs? Alternatively, the authors should consider the possibility that high AKT3 is a hallmark of some –but not all- CMS4 CRC cancers and is not fully specific (considering the high AKT3 expression found in a CMS1 PDX).

The link between AKT3 and EMT, which is supported by correlations, would be much strengthened by examining the expression of EMT markers in CRC cells KO for AKT3, as used in figure 3G.

In the same line, the relationship between AKT3 and FRMD6 is only correlative and it would be worth investigating if the expression of FRMD6 is controlled by AKT3.

Minor points

The authors should indicate how many tumours have been studied in the RNAscope experiment (figure 2D).

Molecular weights on western blot are lacking.

The authors should indicate more precisely which patient data set was used for figures 4F and 4I.

Regarding FRMD6, the authors should refer to the study by Marisa et al. (Plos Med. 2013), which defines FRMD6 as one of the centroid gene for the poor prognosis CRC subtype.

I suggest the authors to discuss the recent article by Varga et al (J Exp Med 2020) showing that AKT-dependent NOTCH3 activation drives tumour progression in a model of mesenchymal colorectal cancer.

Reviewer 2 Report

The authors deals with the expression and role of AKT3 in mesenchymal colorectal cancer cells; they use a strong and well organized experimental plan that initially evaluates the expression of AKT1, AKT2 and AKT3 and then is focused on AKT3 since it is the more expressed.Notably, their samples range from patients (primary cell cultures and PDX) up to cell lines and the results are often matching.

Their conclusions are adequately supported by results even if the overexpression of AKT3 is not always observed (2/7 cell lines; 2/9 PDX) and this may deserve some more discussion.

Some conclusion are speculative, but the strong experimental approach and the high level of used techniques may enforce these speculations.

The introduction is too much long; I suggest shortening it.

Finally, please add the molecular weights on WB

Round 2

Reviewer 1 Report

I have no further comment.